# A decoherence-free subspace in a charge quadrupole qubit

Mark Friesen[1], Joydip Ghosh[1], M.A. Eriksson[1] & S.N. Coppersmith[1]

Quantum computing promises significant speed-up for certain types of computational problems. However, robust implementations of semiconducting qubits must overcome the effects of charge noise that currently limit coherence during gate operations. Here we describe a scheme for protecting solid-state qubits from uniform electric field fluctuations by generalizing the concept of a decoherence-free subspace for spins, and we propose a specific physical implementation: a quadrupole charge qubit formed in a triple quantum dot. The unique design of the quadrupole qubit enables a particularly simple pulse sequence for suppressing the effects of noise during gate operations. Simulations yield gate fidelities 10–1,000 times better than traditional charge qubits, depending on the magnitude of the environmental noise. Our results suggest that any qubit scheme employing Coulomb interactions (for example, encoded spin qubits or two-qubit gates) could benefit from such a quadrupolar design.

[1] Department of Physics, University of Wisconsin-Madison, Madison, Wisconsin 53706, USA. Correspondence and requests for materials should be addressed to M.F. (email: friesen@physics.wisc.edu).

Due to the fragility of quantum information, multiple layers of error suppression will be needed for any scalable implementation of a quantum computer[1]. Active suppression methods include quantum error correction[2] and composite pulse sequences[3–6], while passive strategies include forming decoherence-free subspaces or subsystems[7–11] (DFS), and optimal working points[12] ('sweet spots'). DFS are particularly attractive, because of their minimal overhead requirements. Previous proposals for DFS in quantum dots have focused on spin qubits and the decoherence caused by uniform magnetic field fluctuations, $\delta \mathbf{B}$[13]. For example, if $\mathbf{s}_i$ is a spin operator for the $i$th qubit, then the fluctuation Hamiltonian is given by $\sum_i g \mu_B \delta \mathbf{B} \cdot \mathbf{s}_i$, where $g$ is the Landé $g$-factor and $\mu_B$ is the Bohr magneton. A DFS then corresponds to a logical encoding of the qubit for which both states are equally affected by the fluctuation.

Unfortunately, recent experiments suggest that the dominant noise source for spin qubits is electric field noise[14] ('charge noise'), which rapidly degrades the quantum coherence when the spins are coupled via exchange interactions[15], or effectively transformed into charge qubits[16,17] via spin-to-charge conversion, as in proposals for two-qubit gates[18]. The Hamiltonian for a uniform electric field fluctuation $\delta \mathbf{E}$ acting on an array of charges takes the form $\sum_i e \delta \mathbf{E} \cdot \mathbf{r}_i$, where the position operator for the $i$th electron, $\mathbf{r}_i$, plays an explicit role for charge fluctuations, in contrast to magnetic fluctuations. This position dependence for uniform electric fields is quite different than the case for spins interacting with a uniform (global) magnetic field, suggesting that it could be impossible to form a DFS for charge qubits, or spin qubits that exploit the charge sector. Recent efforts to suppress the effects of charge noise in quantum dots have therefore focused on sweet spots, which typically occur at energy-level anticrossings, and suppress the leading order effects of $\delta \mathbf{E}$[17,19,20].

In this paper, we show that, contrary to expectations, certain dot geometries do support a DFS for charge. To make the discussion concrete, we propose a new type of charge qubit that we call a charge quadrupole (CQ). Our scheme embraces both passive and active noise suppression strategies: symmetries incorporated into the qubit design naturally suppress the effects of uniform electric field fluctuations (passive), while the special form of the Hamiltonian enables dynamical decoupling sequences (active) that are shorter than existing protocols for quantum gate operations. We provide an analytical explanation for the suppression of dephasing within the logical subspace. We also perform simulations that yield substantial improvements in gate fidelities by combining passive and active error suppression, under realistic assumptions about the charge noise. We further propose extensions of the quadrupolar geometry for coupling a charge qubit to a microwave cavity or to another qubit.

## Results

**Decoherence-free subspace.** Before describing the CQ qubit in detail, we first recall a DFS for spins. Three spins can encode a DFS that protects against arbitrary uniform magnetic field fluctuations[21–23], $\delta \mathbf{B}$. The DFS consists of two states with the same values of the total spin along the quantization axis, $S_z = \sum_i s_{zi}$, and of the total spin $S^2 = S_x^2 + S_y^2 + S_z^2$. The DFS has two important properties: first, the difference in the energies of the two qubit states is independent of magnetic field, and second, changing a spin-independent Hamiltonian causes the system to evolve only between the qubit states; non-qubit ('leakage') states cannot be accessed because they are not coupled to the qubit states by the Hamiltonian.

Here we construct a similar arrangement for charge states that protects against uniform electric field variations. All linear superpositions of the logical states must have the same total

charge and also the same centre of mass, so that the contribution to the energy from a uniform field, $\sum_i e \mathbf{E} \cdot \mathbf{r}_i$, is the same for all qubit states. In addition, it is important that the system Hamiltonian does not couple the qubit states to the other states in the full Hilbert space. These conditions are satisfied if the Hamiltonian conserves charge and has an appropriate symmetry: the qubit logical states should have the same total charge and be eigenstates of a symmetry operator with the same eigenvalue. An appropriate candidate geometry is a central dot that is symmetrically coupled to a set of outer dots having the same centre of mass as the centre dot, even under permutation. Analogous to the situation for a spin DFS[1,22], the symmetry constraints cannot be satisfied in a two-dimensional (double dot) code space; the smallest device that can support a charge DFS is a triple dot.

**Charge quadrupole qubit.** Here we consider a linear triple dot geometry, where the symmetry operation is the permutation operator between the outer two dots, $p_{1,3}$, which is equivalent to reflection about the centre. It is convenient to adopt the basis states $\{|C\rangle = |010\rangle, |E\rangle = (|100\rangle + |001\rangle)/\sqrt{2}\}$, where $C$ and $E$ refer to 'center' and 'even.' The resulting $p_{1,3}$ eigenvalue is $+1$, corresponding to even symmetry. The third, orthogonal state, $|L\rangle = (|100\rangle - |001\rangle)/\sqrt{2}$, has eigenvalue $-1$, corresponding to odd symmetry, and generates a dipole that couples to charge fluctuations when superposed with $|E\rangle$. When the symmetry constraint is satisfied, the even and odd manifolds decouple.

The logical charge states of a CQ qubit are protected from uniform electric field fluctuations because their charge distributions have the same centre of mass (in other words, no dipole moment). It is interesting to note that several related systems also propose to use dipole-free geometries, including the zero-detuning sweet spot of a conventional charge qubit[17,24,25], which we analyse below, a three-island transmon qubit[26], and quantum cellular automata[27–29].

We now examine the CQ qubit in more detail. We consider a triple dot with one electron, as illustrated in Fig. 1. The full Hamiltonian in the position basis is given by

$$H_{CQ} = \begin{pmatrix} U_1 & t_A & 0 \\ t_A & U_2 & t_B \\ 0 & t_B & U_3 \end{pmatrix} = \begin{pmatrix} \epsilon_d & t_A & 0 \\ t_A & \epsilon_q & t_B \\ 0 & t_B & -\epsilon_d \end{pmatrix} + \frac{U_1 + U_3}{2}, \tag{1}$$

where $t_A$ and $t_B$ are the tunnelling amplitudes between neighbouring dots, and $U_1$, $U_2$, and $U_3$ are site potentials. We have also defined the dipolar and quadrupolar detuning parameters, $\epsilon_d$ and $\epsilon_q$, as

$$\epsilon_d = (U_1 - U_3)/2 \quad \text{and} \quad \epsilon_q = U_2 - (U_1 + U_3)/2. \tag{2}$$

The eigenvalues of $H_{CQ}$ are plotted as a function of $\epsilon_q$ in Fig. 1, where the lowest and highest energy levels correspond to the logical eigenstates $|\tilde{0}\rangle$ and $|\tilde{1}\rangle$, respectively, and the middle level is the leakage state $|\tilde{L}\rangle$. This ordering is an uncommon but benign feature of the CQ qubit, as shown below. We note that, away from $\epsilon_q = 0$, the eigenstate $|\tilde{L}\rangle$ differs slightly from the basis state $|L\rangle$ due to mixing terms in equation (1). Below, we show that under ideal conditions, the mixing terms are small, so that $|\tilde{L}\rangle \simeq |L\rangle$.

It is instructive to compare $H_{CQ}$ to a conventional, one electron charge qubit formed in a double dot, which we refer to as a charge dipole (CD):

$$H_{CD} = \begin{pmatrix} \epsilon_d/2 & t \\ t & -\epsilon_d/2 \end{pmatrix}. \tag{3}$$

In this case, $\epsilon_d = U_1 - U_2$ is the dipole detuning, and there is no quadrupole detuning. In what follows, we express the detuning

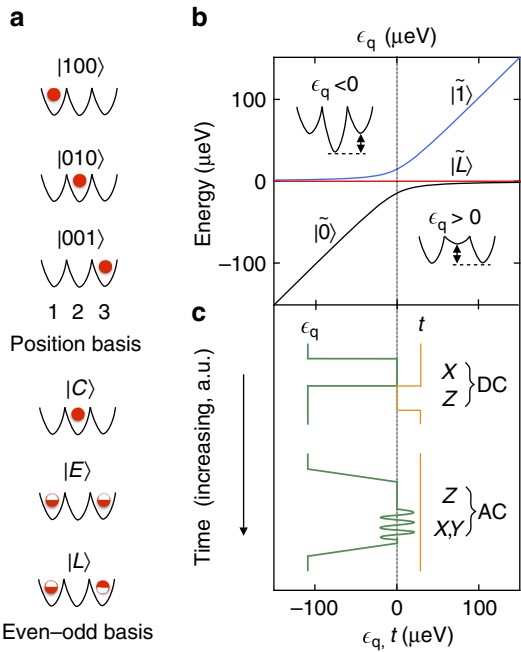

**Figure 1 | Basis states and gate operations of the charge quadrupole qubit.** (**a**) A triple-dot charge qubit can be described in terms of a position basis, corresponding to the single-electron occupations of dots 1, 2, or 3. For a symmetrized triple dot, it is preferable to adopt, instead, an even-odd basis $\{|C\rangle, |E\rangle, |L\rangle\}$, referring to centre, even, and leakage states, respectively. Here, $|C\rangle$ and $|E\rangle$ have even symmetry, $|L\rangle$ has odd symmetry, and the half-filled circles represent average occupations of 1/2. (**b**) The charge quadrupole eigenstates, $|\tilde{0}\rangle$, $|\tilde{1}\rangle$, and $|\tilde{L}\rangle$, obtained by solving the system Hamiltonian, equation (1), as a function of quadrupolar detuning, $\epsilon_q$. Here, we set the tunnel couplings, $t_A = t_B (\equiv t/\sqrt{2}) = 2.5$ GHz and the dipolar detuning, $\epsilon_d = 0$. The insets depict the effect of $\epsilon_q$ on the triple-dot confinement potential. (**c**) A cartoon depiction of microwave (AC) and pulsed (DC) gate sequences useful for qubit manipulation. Initialization and readout are implemented in the far-detuned regime, $\epsilon_q \ll 0$, with $t \gtrsim 0$. In the DC scheme, $\epsilon_q$ is suddenly pulsed to the double sweet spot, $\epsilon_q = 0$. Free evolution then yields an $X$ rotation in the logical basis $\{|C\rangle, |E\rangle\}$. For a $Z$ rotation, $t$ is suddenly pulsed to 0, while $\epsilon_q$ is pulsed away from zero (either positive or negative). In the AC scheme, $t \gtrsim 0$ is held fixed, while an adiabatic ramp of $\epsilon_q$ to its sweet spot leaves the qubit in its logical ground state. $X$ and $Y$ rotations in the rotating frame are implemented by applying resonant microwave bursts with appropriate phases to $\epsilon_q$, centred at the sweet spot. Alternatively, microwaves may be applied to $t$, although we do not consider that possibility here.

parameters in terms of their average ($\bar{\epsilon}$) and fluctuating ($\delta\epsilon$) components. Uniform electric field fluctuations are then associated with $\delta\epsilon_d$, while fluctuations of the field gradient are associated with $\delta\epsilon_q$.

**Charge noise.** It has been shown that phonon decoherence processes can be classified based on multipole moments[30]. Here we consider charge noise decoherence processes for the leading order (dipole and quadrupole) moments in the noise, which by construction we will expect to behave very differently for CD and CQ qubits. Fluctuations in $\delta\epsilon_d$ are dangerous for single-qubit operations since they cause fluctuations of the energy splitting between the qubit levels, $E_{01}$, resulting in phase fluctuations. The success of the DFS depends on our ability to engineer a triple dot in which the dephasing effects of $\delta\epsilon_d$ fluctuations are suppressed.

The next-leading source of fluctuations, $\delta\epsilon_q$, is much weaker, and we show in Supplementary Note 1 that

$$\delta\epsilon_q/\delta\epsilon_d \simeq d/R, \tag{4}$$

where $d$ is the interdot spacing and $R$ is the characteristic distance between the qubit and the charge fluctuators that cause $\delta\epsilon_d$. We also estimate that $d/R \simeq 0.1$ in recent devices used for double dot qubit experiments[17,24]. In Supplementary Note 2, we further show that $\delta\epsilon_d$ is related to the more fundamental noise parameter $\delta E$ (the fluctuating electric field) as $\delta\epsilon_d \propto d\delta E$, so that $\delta\epsilon_q \propto d^2\delta E$. Therefore, the effects of charge noise can be suppressed by making $d$ smaller through engineering, by reducing the lithographic feature size and the interdot spacing. This is one of the key attractions of the quadrupole qubit: it provides a straightforward path for systematically improving the qubit fidelity, by reducing the device size.

The Hamiltonian $H_{CQ}$ has four independently tunable parameters. We now determine the control settings consistent with DFS operation. Our goal is to block diagonalize $H_{CQ}$ so that it decomposes into a two-dimensional (2D) logical subspace, and a 1D leakage space. Any coupling to the leakage space would result in energy-level repulsions as a function of the tuning parameters. We can therefore suppress such coupling by requiring that $\partial E_L/\partial \epsilon_q = \partial E_L/\partial \epsilon_d = 0$, where $E_L$ is the leakage state energy, yielding the desired tunings: $t_A = t_B$ and $\bar{\epsilon}_d = 0$. These are the same conditions obtained by requiring that $[H_{CQ}, p_{1,3}] = 0$, to obtain simultaneous eigenstates of $H_{CQ}$ and $p_{1,3}$. The even-symmetry states $|C\rangle$ and $|E\rangle$ are good choices for basis states in the 2D manifold. With the basis set $\{|C\rangle, |E\rangle, |L\rangle\}$, and the parameter tunings $t_A = t_B$ and $\bar{\epsilon}_d = 0$, we find that $H_{CQ}$ block diagonalizes as desired. (For notational convenience, we define $t/\sqrt{2} \equiv t_A = t_B$.) In the logical subspace $\{|C\rangle, |E\rangle\}$, the reduced Hamiltonian is then given by $H_{CQ,ideal} = (\bar{\epsilon}_q/2)(1 + \sigma_z) + t\sigma_x$, where $\sigma_x$ and $\sigma_z$ are Pauli matrices.

We now compare the effects of fluctuations on the energy levels of CD and CQ qubits. (In the following sections, we explore the effect of fluctuations on gate operations.) The energy splitting of CD qubits is obtained from equation (3) as $E_{01,CD} = \sqrt{\epsilon_d^2 + 4t^2}$. A fluctuation expansion in powers of $\delta\epsilon_d$ yields

$$E_{01,CD} = \sqrt{\bar{\epsilon}_d^2 + 4t^2} + \left[ \frac{\bar{\epsilon}_d}{(\bar{\epsilon}_d^2 + 4t^2)^{1/2}} \right] \delta\epsilon_d$$
$$+ \left[ \frac{2t^2}{(\bar{\epsilon}_d^2 + 4t^2)^{3/2}} \right] \delta\epsilon_d^2 + O[\delta\epsilon_d^3]. \tag{5}$$

The first term in equation (5) indicates that $\bar{\epsilon}_d$ and $t$ are the main control parameters. The second term indicates that the qubit is only protected from fluctuations of $O[\delta\epsilon_d]$ at the sweet spot, $\bar{\epsilon}_d = 0$. In contrast, the CQ qubit has two detuning parameters. In this case, we fix $\bar{\epsilon}_d = 0$ and calculate the energy splitting $E_{01,CQ}$ by writing $\epsilon_d \to \delta\epsilon_d$ and $\epsilon_q \to \bar{\epsilon}_q + \delta\epsilon_q$. Expanding in $\delta\epsilon_d$ and $\delta\epsilon_q$ yields

$$E_{01,CQ} = \sqrt{\bar{\epsilon}_q^2 + 4t^2} + \left[ \frac{\bar{\epsilon}_q}{(\bar{\epsilon}_q^2 + 4t^2)^{1/2}} \right] \delta\epsilon_q$$
$$+ \left[ \frac{\bar{\epsilon}_q^2 + 2t^2}{t^2 (\bar{\epsilon}_q^2 + 4t^2)^{1/2}} \right] \delta\epsilon_d^2 + O[\delta\epsilon_q^2, \delta\epsilon_d^3], \tag{6}$$

where we note that $\delta\epsilon_q^2 \ll \delta\epsilon_d^2$. By construction, $E_{01,CQ}$ has no terms of $O[\delta\epsilon_d]$ when $\bar{\epsilon}_d = 0$. Moreover, we see that fluctuations of $O[\delta\epsilon_q]$ vanish when $\bar{\epsilon}_q = 0$. Hence, $\bar{\epsilon}_d = \bar{\epsilon}_q = 0$ represents a double sweet spot. Since $\delta\epsilon_q \ll \delta\epsilon_d$, dephasing is minimized when

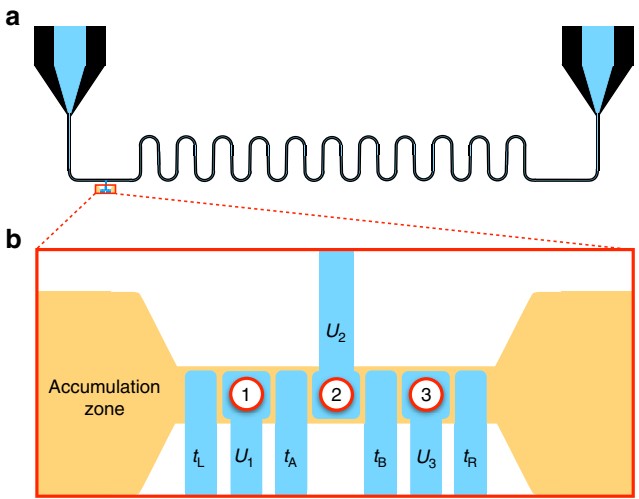

**a**

**b**

**Figure 2 | Preserving the symmetry of the quadrupole qubit with external couplings.** Schematic of the geometry of (**a**) a microwave stripline resonator coupled to (**b**) a quadrupole qubit. The stripline geometry shown is similar to those suggested in refs 34,54–56. Both accumulation-mode gates that control the dot occupations (with local potentials labelled $U_1$, $U_2$, and $U_3$) and depletion-mode gates that control the tunnel couplings (labelled $t_L$, $t_A$, $t_B$, and $t_R$) are included here (see refs 48,60–62 for a discussion). The corresponding dots are labelled 1, 2, and 3. The coupling occurs through the middle gate 2, which is connected to the resonator. The qubit can be coupled, similarly, to other qubits or charge sensors.

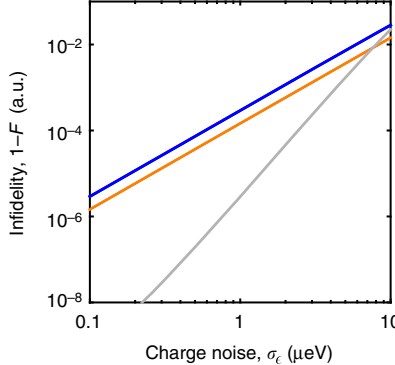

**Figure 3 | Simulated process fidelities for charge qubit gate operations.** Simulations of bare $X_\pi$ rotations and composite pulse sequences, $\tilde{X}_\pi$, for charge dipole (CD) and charge quadrupole (CQ) qubits are performed in the presence of dipolar ($\delta\epsilon_d$), and quadrupolar ($\delta\epsilon_q = \delta\epsilon_d/40$) detuning fluctuations. Plots show the infidelity ($=1-$fidelity) as a function of the s.d. $\sigma_\epsilon$ of $\delta\epsilon_d$. (The simulations, charge noise averages, and process fidelity calculations are described in Methods.) Here, the blue and orange curves correspond to bare, single-pulse $X_\pi$ rotations of CQ and CD qubits, respectively. The infidelity follows the same scaling in both cases, even though it arises from different mechanisms: pure dephasing for the CD qubit versus leakage for the CQ qubit. The grey curve corresponds to a composite, three-pulse sequence, $\tilde{X}_\pi \equiv Z_{2\pi}X_{3\pi}Z_{-2\pi}$, which removes the leading order $\delta\epsilon_d$ noise in the CQ qubit; no comparable sequence exists for CD qubits. The simple form of the CQ pulse sequence derives from the quadrupole geometry, which transfers some of the overhead for noise protection from the control pulse sequence to the qubit hardware. All simulations assume the same tunnel couplings ($t$ for the CD qubit; $t_{A,B}$ for the CQ qubit) of 10 GHz.

we set $\bar{\epsilon}_d = 0$ and adopt $\bar{\epsilon}_q$ and $t$ as the control parameters for CQ gate operations. Although $\delta\epsilon_d$ does not appear at linear order in $E_{01,CQ}$, its main effect is to cause leakage for CQ qubits, rather than dephasing—a point that we return to below. For both CD and CQ qubits, we note that increasing the tunnel coupling $t$ also suppresses the energy fluctuations, particularly near the sweet spot; this is consistent with recent results in a resonantly gated three-electron exchange-only qubit[31,32].

**Pulsed gates for the CQ qubit.** Here we investigate pulsed (DC) gates, assuming that $\bar{\epsilon}_q$ and $t$ can be independently tuned and set to zero. We perform rotations of angle $\alpha$ around the $\hat{x}$ axis of the Bloch sphere ($X_\alpha$) by setting $\bar{\epsilon}_q = 0$ and $t = t_x > 0$. Rotations of angle $\beta$ around the $\hat{z}$ axis ($Z_\beta$) are achieved by setting $\epsilon_q = \epsilon_z \neq 0$ and $t = 0$. Readout is performed by measuring the charge occupation of the centre dot. In fact, all external couplings to initialization and readout circuits or to other qubits should be made through the centre dot, to preserve the symmetries of the qubit, as illustrated in Fig. 2.

We compare gate operations in CQ qubits to those of CD qubits via simulations that include quasistatic charge noise. Results for the infidelities of simple ('bare') $X_\pi$ rotations for CQ and CD qubits are presented in Fig. 3 (blue and orange curves, respectively), as functions of the s.d. of charge noise fluctuations (see Methods section for details). Both curves follow the same scaling behaviour, which can be explained as follows. The effect of quasistatic noise, $\delta\epsilon_d$, may be regarded as a gating error that induces underrotations, overrotations, or rotations about a misoriented axis. For CD qubits, such errors occur within the logical Hilbert space, while for CQ qubits, the misrotation occurs primarily to the leakage state. Defining the state fidelity as $F_s = |\langle \psi_{actual} | \psi_{ideal} \rangle|^2$, where $|\psi_{actual}\rangle$ is the actual final state and $|\psi_{ideal}\rangle$ is the target state, the noise drives $F_s < 1$. For either qubit,

the resulting infidelity of noisy rotations scales as $(1 - F_s) \propto \delta\epsilon_d^2$. For example, the probability of a CQ qubit being projected onto its leakage state is $|\langle \tilde{L} | C \rangle|^2 = \delta\epsilon_d^2 / (\delta\epsilon_d^2 + t^2) \sim \delta\epsilon_d^2 / t^2$.

**Composite pulse sequences.** Fortunately, time evolution remains largely coherent throughout a gate operation, so that special pulse sequences can be used to undo the leakage and suppress the errors[5,6,33]. In ref. 33, three-pulse sequences were constructed for the CQ qubit, following the same control constraints indicated in Fig. 1c, which are experimentally motivated: the control parameters $\epsilon_q$ and $t$ can be pulsed independently, but not simultaneously, between zero and a finite value. There it was shown that special values of the control parameters can be used to cancel out the leading order effect of $\delta\epsilon_d$ noise, yielding a universal set of low-leakage, single-qubit gate operations. In Fig. 3, we compare one such composite sequence for the CQ qubit, $\tilde{X}_\pi \equiv Z_{2\pi}X_{3\pi}Z_{-2\pi}$, to bare, single-step sequences for $X_\pi$ rotations in both CD and CQ qubits. The results show that significant benefits can be achieved with composite sequences: for charge noise levels consistent with recent experiments[19,34], fidelity improvements are in the range of 10–1,000.

While noise-cancelling pulse sequences have been proposed for quantum dot spin qubits[5,6], they are significantly more complex than the three-pulse sequence used in Fig. 3. Those sequences are constructed by inserting identity operations into the pulse sequence and assuming a continuous range of rotation axes in some plane of the Bloch sphere. The constraints assumed above, where $\epsilon_q$ and $t$ are not varied simultaneously, yield bare $X$ and $Z$ rotations, but no continuous range of rotation axes. Under such conditions, no three-pulse sequence exists that can cancel out leading-order $\delta\epsilon_d$ noise in CD qubits. By relaxing these

constraints to allow simultaneous tuning of $\epsilon_q$ and $t$—a challenging but potentially achievable goal—it becomes possible to construct a five-step sequence to cancel out the leading-order noise in CD qubits. Thus, the three-step sequence described above for CQ qubits is truly 'minimal,' in the sense that it has the same level of complexity as a conventional spin-echo sequence, which has been shown to be effective for preserving the coherence of a CD qubit[25].

**Microwave-driven gates**. While it is necessary to move away from the sweet spot to perform certain microwave-driven (AC) gate operations, it is possible to centre the AC signal at the sweet spot, as indicated in Fig. 1c, which improves the coherence of gate operations. Below, we analyse the AC gate sequence shown in Fig. 1c. Here initialization and readout are performed in the far-detuned regime. However, we now consider an adiabatic ramp to the sweet spot, so that the working point $\bar{\epsilon}_q = 0$ defines the quantization axis $\hat{z}$ in the laboratory frame. For AC gates, one typically moves to the frame rotating at the qubit frequency, where $X$ and $Y$ rotations are obtained by driving the appropriate detuning parameter (dipolar for a CD qubit, quadrupolar for a CQ qubit), with the appropriate phase, at the resonance frequency $\nu = E_{01}/h$.

In the rotating frame, the primary decoherence mechanism during $X$ or $Y$ rotations is longitudinal, with the corresponding decay time $T_{1\rho}$ (ref. 35). In this case, the charge noise environment is nearly Markovian, so that, on resonance, it is sufficient to use Bloch-Redfield theory, giving[36]

$$1/T_{1\rho} = 2S_z(\epsilon_{ac}/\hbar) \\ + S_x([\epsilon_{ac} + 2t]/\hbar) + S_x([\epsilon_{ac} - 2t]/\hbar), \qquad (7)$$

where $\epsilon_{ac}$ is the amplitude of the resonant drive, and $S_z(\omega)$ and $S_x(\omega)$ are the longitudinal and transverse noise spectral densities in the lab frame, respectively. These functions describe the noise in the detuning parameters used to drive the rotations ($\epsilon_d$ for CD qubits, or $\epsilon_q$ for CQ qubits). In the weak driving regime, $\epsilon_{ac} \ll 2t$, the term $2S_z(\epsilon_{ac}/\hbar)$ would normally dominate equation (7) because $S_{x,z}(\omega) \propto 1/\omega$ for charge noise. However, at the sweet spot, the $\epsilon$ noise for either type of qubit is orthogonal to the quantization axis, so that $S_z(\omega) = 0$. The other terms in equation (7) are relatively small, since their arguments are large.

We can compare $T_{1\rho}$ for CD and CQ qubits by assuming that the noise terms, $\delta\epsilon_d$ and $\delta\epsilon_q$, both arise from the same charge fluctuators. In this case, the ratio of their amplitudes, $\delta\epsilon_q/\delta\epsilon_d$, is independent of the frequency and the decoherence rates for resonant $X$ and $Y$ rotations in a CQ qubit are suppressed by this same ratio, as compared to a CD qubit. After applying simple pulse sequences to suppress the leakage, CQ qubits are therefore protected from the dominant ($O[\delta\epsilon_d]$) noise source for all rotation axes, for both pulsed and resonant gates, while CD qubits require a more complex correction scheme.

**Spin quadrupole qubits**. Up to this point, we have focused on charge qubits. However, quadrupolar geometries can also be used to protect logical spin qubits from dipolar detuning fluctuations. For example, the standard two-electron singlet-triplet ($S$-$T$) qubit formed in a double quantum dot[37,38] is not protected from dipolar detuning fluctuations during implementation of an exchange gate. But a singlet-triplet qubit formed in a triple dot could be protected by tuning the device, symmetrically, to one of the charging transitions, $(1, 0, 1) - (1/2, 1, 1/2)$ or $(0, 2, 0) - (1/2, 1, 1/2)$. Here the delocalized states with half-filled superpositions are analogous to those shown in Fig. 1. The magnitudes of the local Overhauser fields on dots 1 and 3 should be equalized for $S$-$T$ qubits, to enforce the symmetry requirements and

suppress leakage out of the logical subspace. We note that a different type of symmetric sweet spot was recently employed for a singlet-triplet qubit in a double-dot geometry[39,40]. In those experiments, the resonant pulse was applied to the tunnel coupling, as suggested in ref. 41, while the detuning parameter was set to a sweet spot.

Three-electron logical spins, such as the quantum dot hybrid[19,42–46] or exchange-only[21,31,32,47,48] qubits, can also be implemented using a quadrupolar triple dot. In this case, we must work at one of the charging transitions $(1, 1, 1) - (3/2, 0, 3/2)$, $(1, 1, 1) - (1/2, 2, 1/2)$, or $(0, 3, 0) - (1/2, 2, 1/2)$. When the qubit basis involves singlet- and triplet-like spin states[42,43], localized in dots 1 or 3 (for example, at the $(1, 1, 1) - (3/2, 0, 3/2)$ transition), the $S$-$T$ splittings in those dots should be equalized. We note that measuring exchange-only qubits, or performing capacitive two-qubit gate operations, requires accessing the charge sector of those devices. The conventional charging transition used for this purpose is $(1, 1, 1) - (2, 0, 1)$[47], which is not protected from $\delta\epsilon_d$ fluctuations. A symmetric quadrupolar geometry could therefore benefit such operations.

**External couplings and two qubit gates**. Two main types of couplings have been proposed for two-qubit gates in quantum dot qubits: classical electrostatic (capacitive) interactions[18] or quantum exchange interactions[15]. We only consider capacitive couplings here, since exchange couplings require the dots to be in very close proximity. Capacitive couplings mediated by qubit proximity or floating top gates[49] are convenient for quadrupole qubits, provided that the device symmetries are preserved during gate operations. This suggests that the coupling should occur through the gate above the middle dot. Readout and charge-to-photon interconversions should also be performed in the same way.

Capacitive two-qubit interactions, which yield an effective coupling of form $J\sigma_{z1}\sigma_{z2}$ in the basis $\{|C\rangle, |E\rangle\}_1 \otimes \{|C\rangle, |E\rangle\}_2$, have previously been demonstrated in CD qubits[50] and in logical spin qubits[51]. Here $J(\epsilon_{q1}, \epsilon_{q2})$ represents the capacitive dipole–dipole coupling, and the indices 1 and 2 refer to the interacting qubits. One advantage of this coupling is that no new leakage states are incurred, beyond the single-qubit states $|L\rangle_1$ and $|L\rangle_2$, in contrast with two-qubit gates in some other DFS[21].

We now describe a simple protocol for nonadiabatic, pulsed two-qubit gate operations for CQ qubits, based on schemes developed for Cooper-pair boxes[52], which are superconducting versions of the CD qubit. (AC gating schemes based on state-dependent resonant frequencies are also candidates for two-qubit operations[53], although we do not consider them here.) Our DC scheme can be viewed as a shift of the degeneracy point $\epsilon_{q2} = 0$ of qubit 2, depending on the state of qubit 1. The qubits are first prepared in their ground states in the far-detuned regime, yielding $|\tilde{0}\tilde{0}\rangle$. Qubit 2 is then pulsed to its degeneracy point, where free evolution yields an $X_\pi$ rotation to state $|\tilde{0}\tilde{1}\rangle$. On the other hand, if an $X_\pi$ rotation is first applied to qubit 1, so the system is in state $|\tilde{1}\tilde{0}\rangle$, there will be an effective shift in $\epsilon_{q2}$ due to the interaction term. Now when $\epsilon_{q2}$ is pulsed, it does not reach its degeneracy point. In this case, no $X_\pi$ is implemented on qubit 2, and the system remains in state $|\tilde{1}\tilde{0}\rangle$. The net result is a controlled (C)-NOT gate. We note that since the qubits spend most of their time away from sweet spots in this protocol, the special noise protection afforded by CQ qubits should significantly improve their coherence.

Other types of external couplings are also possible. For example, a microwave stripline resonator could potentially enable two-qubit couplings, readout, and charge-to-photon conversions by techniques described in refs 34,54–56, when coupled to a CQ

qubit, as illustrated in Fig. 2. Qubit-resonator coupling strengths in the range $g = 5$–$50$ MHz have been reported for cavity quantum electrodynamic (cQED) systems employing CD qubits[34,54–56]. Strong coupling has been achieved in such devices[34], but it is challenging[57], due to short CD coherence times of order 1 ns. Achieving strong coupling requires that both $g/\Gamma_q \gg 1$ and $g/\Gamma_s \gg 1$, where $\Gamma_q \sim 1/T_{1\rho}$ is the main decoherence rate for the qubit, and $\Gamma_s$ is the decoherence rate for the superconducting stripline. We expect that $g$ for CQ qubits should be similar to CD qubits, while $\Gamma_q$ should be reduced by a factor of $\sim 10$, so $g/\Gamma_q$ should increase by a factor of $\sim 10$, which would mitigate the difficulties in achieving strong coupling. It should also be possible to couple microwave striplines to quadrupolar spin qubits, using spin-to-charge conversion[58].

## Discussion

In conclusion, we have shown that charge qubit dephasing can be suppressed by employing a quadrupolar geometry, because the quadrupolar detuning fluctuations are much weaker than dipolar fluctuations. On the other hand, the quadrupolar detuning parameter $\epsilon_q$ is readily controlled by applying voltages to the top gates, and we expect gate times for CQ qubits to be just as fast as CD qubits. Since dephasing is suppressed for CQ qubits while gate times are unchanged, we expect noise suppression techniques to be more effective for CQ qubits than CD qubits. We have confirmed this by simulating minimal composite pulse sequences, designed to cancel out the effects of leakage. This is a promising result for charge qubits because the fidelities of pulsed[17] and microwave[25] gating schemes are not currently high enough to enable error correction during gate operations. We have also shown that the coherence properties of CQ qubits improve as the devices shrink, and we expect future generations of small CQ qubits to achieve very high gate fidelities. We have further shown that logical spin qubits in quantum dots should benefit from a quadrupolar geometry. We expect a prominent application for quadrupolar qubits to be cQED, where improvements in coherence properties could enhance strong coupling.

## Methods

**Simulations and fidelity calculations.** Gate operations on CQ and CD qubits were simulated using standard numerical techniques to solve $i\hbar\dot\rho = [H, \rho]$, where $\rho$ is the 2D (3D) density matrix for the CD (CQ) qubit, defined by Hamiltonian $H = H_{CD}$ ($H_{CQ}$). For the simulations shown in Fig. 3, $H$ implements either a simple $X_\pi$ rotation, or a composite rotation $\tilde X_\pi$, as defined in the main text and Supplementary Note 3. Process tomography is performed using the Choi-Jamiolkowski representation[59] for process $\mathcal{E}(\rho)$, defined by the evolution of $i\hbar\dot\rho_{\mathcal{E}} = [I \otimes H, \rho_{\mathcal{E}}]$, where $I$ is the identity matrix of an ancilla qubit with the same dimensions as $H$. Here, the initial Jamiolkowski state is given by $\rho_{\mathcal{E}}(0) = |\Phi\rangle\langle\Phi|$, where $|\Phi\rangle = (|00\rangle + |11\rangle)/\sqrt{2}$ for the CD qubit, and $|\Phi\rangle = (|CC\rangle + |EE\rangle)/\sqrt{2}$ for the CQ qubit. Formed in this way, $\rho_{\mathcal{E}}$ is equivalent to the standard $\chi$ matrix[59], and we compute $F = \mathrm{Tr}[\chi_{\mathrm{ideal}}\chi_{\mathrm{actual}}]$, where $\chi_{\mathrm{ideal}}$ is obtained by setting $\delta\epsilon_d = \delta\epsilon_q = 0$.

**Charge noise averages.** Averages over quasistatic charge noise were performed using a gaussian probability distribution $P$ sampled from 41 equally spaced points in the range $\delta\epsilon_d \in [-6\sigma_\epsilon, 6\sigma_\epsilon]$, where

$$P(\delta\epsilon_d) = \frac{1}{\sqrt{2\pi\sigma_\epsilon^2}}\exp\left(-\frac{\delta\epsilon_d^2}{2\sigma_\epsilon^2}\right), \tag{8}$$

and $\sigma_\epsilon$ is the s.d. of the distribution.

**Data availability.** Data sharing not applicable to this article as no data sets were generated or analysed during the current study. The simulation results reported in Fig. 3 may be obtained identically by following the computational scheme described above.

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

## Acknowledgements

We thank Robin Blume-Kohout, Adam Frees, and John Gamble for helpful conversations and information. This work was supported by ARO under award no. W911NF-12-0607, and by NSF under award no. PHY-1104660. The authors would also like to acknowledge support from the Vannevar Bush Faculty Fellowship program sponsored by the Basic Research Office of the Assistant Secretary of Defense for Research and Engineering and funded by the Office of Naval Research through grant no. N00014-15-1-0029.

## Author contributions

M.F., S.N.C. and M.A.E. designed the qubit. J.G. designed the leakage suppression pulse sequence. M.F. performed the numerical simulations. All authors analysed the results and prepared the manuscript.

## Additional information

**Competing interests:** M.F., M.A.E. and S.N.C. are co-inventors on a patent application related to this manuscript. The remaining authors declare no competing financial interests.

**Publisher's note**: 

