## [Peer Review File · Nature Communications]

Reviewers' comments:

Reviewer #1 (Remarks to the Author):

This work introduces a charge qubit design that alleviates charge fluctuation susceptibility. This is achieved by constructing a symmetric arrangement of three tunnel coupled quantum dots such that its low energy subspace with a single electron is spanned by three states two of which have vanishing dipole matrix elements, i.e. are only coupled via a weaker quadrupole moment. A logical qubit can be encoded in this two-dimensional subspace. This idea is interesting and in the context of charge-based qd qubits, seems to be original.

It seems that his work has some relation to previous works in other fields. In particular I would like the authors to comment on similarities and differences with PRL 106, 030502 (2011) (Gambetta et al.). They do not need to cite this work if they don't think it is relevant.

The protection reportedly works when the inter-QD distance is small compared with the distance of the QDs to charge fluctuators. Although the authors claim this can be guaranteed by making the dots smaller it is not clear to the referee that this is sufficient. Aren't the fluctuator positions random ?

The manuscript is well written and the idea is sufficiently interesting to warrant publication in Nat. Comm.

Best regards,
Simon Nigg

Reviewer #2 (Remarks to the Author):

The paper "decoherence-free subspace for charge: the quadrupole qubit" by Friesen et al. is a well written and well referenced proposal for an interesting and novel way to overcome the problem of charge noise in quantum dot based charge qubits. The authors first argue that such noise already arises when spin (not charge) qubits are coupled via exchange interactions, or effectively transformed into charge qubits via spin-to-charge conversions, and is nowadays considered to be the primary source of decoherence in quantum dot based spin qubits.

Charge qubits are notoriously susceptible to uniform field fluctuations. The key idea in this work is to design a new type of decoherence-free subspace qubit they call a "charge quadrupole" (CQ) that couples primarily to gradient field fluctuations, as opposed to uniform field fluctuations. This qubit comprises three physical quantum dots arranged linearly occupied by a single electron. The three position basis states of the electron are recombined into the pair $|010\rangle$ and $(|100\rangle + |001\rangle)/\sqrt{2}$, which form the CQ qubit, while the third orthogonal state is a leakage state. The authors show, using a back of the envelope style analysis that starts from the 3×3 Hamiltonian of the CQ Hamiltonian in the position basis, that the CQ qubit is well protected against dipolar charge fluctuations. The protection improves as the inter-dot distance is shrunk, in the sense that the ratio of the dipolar charge fluctuation to the quadrupolar charge fluctuation scales as the inter-dot distance (divided by characteristic separation between the fluctuator and the quantum dot). The linear arrangement is shown to be necessary to compensate for the fact that uniform electric field fluctuations can also generate quadrupolar detuning fluctuations.

The ideas are extended to protect logical spin qubits, such as the singlet-triplet qubit, from dipolar detuning fluctuations

The work shows a very detailed level of familiarity with experimental aspects of charge and spin qubits in quantum dots, which is to be expected from this team of authors. Many technical aspects

are well accounted for, and overall the authors make a compelling case that this new type of qubit will be superior to current designs, including in terms of T1 times. This is an important conclusion that can drive the field of quantum dots qubits into a new and promising direction.

My main concerns are regarding the discussion of logic gates.

- The discussion of single logical qubit gates is too brief and somewhat short on technical details. It is difficult to evaluate its validity and simulations illustrating the fidelity as a function of various control and noise parameters would have helped.

- While there is a short discussion of two qubit gates towards the end of the paper (capacitive using a microwave stripline resonator), there is virtually no analysis of this aspect. E.g., what is the form of the Hamiltonian, what is the effective Hilbert space and what about leakage, how should the interactions be switched in order to realize a two-qubit gate between the logical qubits? The coupling of logical qubits should be carefully addressed in order to complete the proposal.

To summarize, this is strong work, though the analysis is rather simplistic. I'd like to see this work published in Nature Communications since I believe the basic idea of the charge quadrupole qubit can move the field of quantum dot qubits forward in a new and promising direction. However, the analysis can certainly benefit from being made more rigorous beyond the back-of-the-envelope style (this can go into the appendix), figures should be added to illustrate gate fidelities, and the coupling of logical qubits should be carefully addressed before the paper can be published.

Reviewer #1 (Remarks to the Author):

This work introduces a charge qubit design that alleviates charge fluctuation susceptibility. This is achieved by constructing a symmetric arrangement of three tunnel coupled quantum dots such that its low energy subspace with a single electron is spanned by three states two of which have vanishing dipole matrix elements, i.e. are only coupled via a weaker quadrupole moment. A logical qubit can be encoded in this two-dimensional subspace. This idea is interesting and in the context of charge-based qd qubits, seems to be original.

It seems that his work has some relation to previous works in other fields. In particular I would like the authors to comment on similarities and differences with PRL 106, 030502 (2011) (Gambetta et al.). They do not need to cite this work if they don't think it is relevant.

We thank the referee for bringing this interesting paper to our attention. Although the Purcell noise mechanism is most relevant in a superconducting context, the proposed solution geometry clearly has similarities to our quadrupole qubit. We therefore cite the article now, as ref. 26. Since many details of our proposal pertain specifically to quantum dots, the paragraph containing the citation on p.2 has been edited to clarify the complementary nature of the two schemes.

The protection reportedly works when the inter-QD distance is small compared with the distance of the QDs to charge fluctuators. Although the authors claim this can be guaranteed by making the dots smaller it is not clear to the referee that this is sufficient. Aren't the fluctuator positions random ?

The referee is correct in noting that the fluctuator positions are random. We did not mean to imply that small dots would guarantee protection, only that the likelihood of protection would be high. We have modified the discussion in the final paragraph of Supplementary Note 1 to make this point more clearly.

The manuscript is well written and the idea is sufficiently interesting to warrant publication in Nat. Comm.

We are very grateful for this assessment.

Reviewer #2 (Remarks to the Author):

The paper "decoherence-free subspace for charge: the quadrupole qubit" by Friesen et al. is a well written and well referenced proposal for an interesting and novel way to overcome the problem of charge noise in quantum dot based charge qubits. The authors first argue that such noise already arises when spin (not charge) qubits are coupled via exchange interactions, or effectively transformed into charge qubits via spin-to-charge conversions, and is nowadays considered to be the primary source of decoherence in quantum dot based spin qubits.

Charge qubits are notoriously susceptible to uniform field fluctuations. The key idea in this work is to design a new type of decoherence-free subspace qubit they call a "charge quadrupole" (CQ) that couples primarily to gradient field fluctuations, as opposed to uniform field fluctuations. This qubit comprises three physical quantum dots arranged linearly occupied by a single electron. The three position basis states of the electron are recombined into the pair $|010\rangle$ and $(|100\rangle+|001\rangle)/\sqrt{2}$, which form the CQ qubit, while the third orthogonal state is a leakage state. The authors show, using a back of the envelope style analysis that starts from the 3×3 Hamiltonian of the CQ Hamiltonian in the position basis, that the CQ qubit is well protected against dipolar charge fluctuations. The protection improves as the inter-dot distance is shrunk, in the sense that the ratio of the dipolar charge fluctuation to the quadrupolar charge fluctuation scales as the inter-dot distance (divided by characteristic separation between the fluctuator and the quantum dot). The linear arrangement is shown to be necessary to compensate for the fact that uniform electric field fluctuations can also generate quadrupolar detuning fluctuations.

The ideas are extended to protect logical spin qubits, such as the singlet-triplet qubit, from dipolar detuning fluctuations

The work shows a very detailed level of familiarity with experimental aspects of charge and spin qubits in quantum dots, which is to be expected from this team of authors. Many technical aspects are well accounted for, and overall the authors make a compelling case that this new type of qubit will be superior to current designs, including in terms of T_1 times. This is an important conclusion that can drive the field of quantum dots qubits into a new and promising direction.

We are very grateful for these kind comments.

My main concerns are regarding the discussion of logic gates.

- The discussion of single logical qubit gates is too brief and somewhat short on technical details. It is difficult to evaluate its validity and simulations illustrating the fidelity as a function of various control and noise parameters would have helped.

We thank the referee for this very helpful suggestion. We have now performed a wide range of simulations to understand and characterize the operation of the quadrupole qubit. In a new section of the manuscript entitled “DC gates for the CQ qubit,” we provide an extended discussion of single-qubit gates, such as those indicated in Fig. 1(c), with simulation highlights reported in Fig. 3.

In our simulations, we found that the fidelities of bare gates are insufficient for characterizing the key differences between charge quadrupole (CQ) and dipole (CD) qubits, as indicated by the blue and red curves in Fig. 3. Fortunately, a member of our group (Dr. Joydip Ghosh) was already working on this problem (see ref. 33), and was able to help clarify the issue. We now understand that the CQ design enables a tradeoff between the “software” used for noise suppression (e.g., dynamical decoupling pulse sequences) and the hardware (the quadrupole design), which allows us to implement highly efficient pulse sequences that greatly improve the gate fidelity. These conclusions are discussed in another new section entitled “Composite pulse sequences.” Because of these contributions, Dr. Ghosh is a co-author on the current version of the manuscript.

- While there is a short discussion of two qubit gates towards the end of the paper (capacitive using a microwave stripline resonator), there is virtually no analysis of this aspect. E.g., what is the form of the Hamiltonian, what is the effective Hilbert space and what about leakage, how should the interactions be switched in order to realize a two-qubit gate between the logical qubits? The coupling of logical qubits should be carefully addressed in order to complete the proposal.

This comment is also very helpful. To address it, we have added a third new section to the manuscript entitled “External coupling and two-qubit gates.” The gating scheme is based on older schemes developed for Cooper-pair boxes, which have many similarities to dipole charge qubits in quantum dots. The new discussion includes the form of the coupling Hamiltonian, the switching scheme for a CNOT gate, and additional comments. We note that the capacitive coupling scheme does not introduce any new leakage states beyond those already present for single qubits.

To summarize, this is strong work, though the analysis is rather simplistic. I'd like to see this work published in Nature Communications since I believe the basic idea of the charge quadrupole qubit can move the field of quantum dot qubits forward in a new and promising direction. However, the analysis can certainly benefit from being made more rigorous beyond the back-of-the-envelope style (this can go into

the appendix), figures should be added to illustrate gate fidelities, and the coupling of logical qubits should be carefully addressed before the paper can be published.

Again, we thank both referees for their encouraging comments and helpful suggestions. We feel that we have carefully addressed all of their concerns. Additionally, we have modified the formatting of the manuscript to match the style of Nature Communications. In order to incorporate the new sections cohesively, we have made numerous, smaller changes throughout the manuscript. We have also moved the third Supplemental Note from the previous version to 'Methods' (after significant updates), and replaced it with a new Supplemental Note providing technical details about the pulse sequence used in Fig. 3.

Both reviewers provided confidential comments to the editor, supporting publication of the manuscript.